

# Potential for agricultural production on disturbed soils mined for apatite using legumes and beneficial microbes

Rebecca Swift[1], Liza Parkinson[1], Thomas J. Edwards, [1] Regina Carr[1], Jen McComb[1], Graham W. O'Hara[1], Giles E. St. John Hardy[1], Lambert Bräu[2], John Howieson[1]

[1]School of Veterinary and Life Sciences, Murdoch University, Murdoch, Western Australia
[2]Centre for Regional and Rural Futures, Deakin University, Burwood, Victoria, Australia
*Correspondence to*: John Howieson (J.Howieson@murdoch.edu.au)

**Abstract.** Christmas Island has been mined for rock phosphate for over 100 years, and as mining will finish in the next few decades there is a need to develop alternative economies on the island, such as high value crop production. However, to conserve the unique flora and fauna on the island, only land previously mined will be considered for this purpose. As these soils have been severely perturbed by mining, strategies to improve soil quality parameters need to be undertaken before plant based industries can be considered. For instance, legumes and beneficial microbes have demonstrated a positive role in the remediation of degraded soils. Therefore, this study aimed to establish the scientific basis upon which agriculture can effectively be developed on soils post phosphate mining. Six legume species (*Glycine max* (Soybean), *Vigna radiata* (Mungbean), *V. unguiculata* (Cowpea), *Phaseolus vulgaris* (Navybean), *Cajanus cajan* (Pigeon pea), and *Lablab purpureus* (Lablab) were sown onto a two ha rehabilitated site that had previously been mined for rock phosphate. The soil had a pH of 7.0, and was high in P but low in Bo, Cu, K, Mg, N and S and had low organic C. The legumes were inoculated with their respective rhizobial inoculant or co-inoculated with the rhizobia and a plant growth promoting bacteria (PGPB) at three different fertilizer rates (nil, a low rate, and five times the low rate). With the exception of *P. vulgaris*, all the legume species survived. The application of fertilizer was essential for maximum biomass yields 18 weeks after sowing, however the lower fertilizer rate was sufficient to obtain maximum yields for some cultivars. The PGPB increased yields and nodulation of some of the legumes at different fertilizer levels. Although the legumes (except *P. vulgaris*) grew in the Christmas Island environment, selection of appropriate legume cultivars and inoculants plus optimization of the fertilizer regime is required for reliable agricultural productivity on the island.

Key words: Christmas Island; rhizobia; Plant Growth Promoting Bacteria; rehabilitation; rock phosphate

## 1 Introduction

When phosphate deposits are exhausted and mining ceases, there may be severe impacts on local economies. During the mining process, the natural vegetation and soil have frequently been removed to pinnacles of bedrock, and in many instances the replacement soils are low in some nutrients and organic matter but often high in heavy metals. In various countries there



have also been problems in water availability and erosion, weed invasion and fire. In arid areas such as north west Queensland, restoration to natural vegetation is the goal (Gillespie and Erskine, 2012) while in more temperate environments such as Florida, phosphate mines have been returned to wetlands, natural forest or grazing land (Hanlon et al., 1996; Brown, 2005). In many cases restoration to natural vegetation is not possible. In Morocco, phosphate mined land has been successfully planted

with citrus and olives as well as forest species (United Nations Environment Program, 2001), but rehabilitation to agricultural cropping is uncommon.

Post mining reclamation of the land needs to consider not only if the land is suitable for restoration to its natural vegetation or to be developed for agriculture, but also the economic requirements of local residents. When phosphorus reserves are depleted,

the loss of income can have serious effects on the local economy.  For instance, 80 % of the island of Nauru, located in the central Pacific, was mined for phosphorus until 2000. The mined land is largely unsuitable for crops or grazing and although efforts are continuing to develop agricultural enterprises (Fa'anunu, 2012) this small nation at present has crippling economic and health issues (Gowdy and McDaniel, 1999). Similarly, on the small island of Banaba restoration of the phosphate mined land to productivity appears very difficult resulting in relocation of the population.

Actions are now being taken to prevent Christmas Island experiencing similar problems to Nauru when rock phosphate mining operations cease in 2034 (King and Snowdon, 2013). Christmas Island is a remote tropical island in the Indian Ocean, 380 km south of Jakarta (Indonesia) (Beeton et al., 2010; National Archives of Australia, 2013). The island crowns an ancient volcano that rises 5000 m above the sea floor (Director of National Parks, 2012) with a cap consisting of alternating carbonic and

volcanic rocks extending to a height of 361 m above sea level (Trueman, 1965). The climate is tropical with a wet season from November to May and an average annual rainfall of 2034 mm (Bureau of Meteorology, 2013). The rainforest is conserved in a National Park which covers 63 % of the island, and is supported by a levy (currently $1.2m pa) from Phosphorus Resources Limited.  On the 22 % (3,000 ha) of the island that has been mined for apatite, if soil quality can be improved, the rehabilitated mined areas will provide an opportunity to develop alternative industries to replace mining for the economic future of the

island.  Unlike Nauru, Christmas Island was uninhabited before the discovery of phosphate so there is no tradition of self-sustaining food production. Most of the food for Christmas Island is imported and expensive, and large scale commercial agricultural cropping has never been attempted.

Christmas Island soils are currently a constraint to agriculture. Most are phosphatic, derived from limestone or basalt with a

pH 7.0-8.0 (Director of National Parks, 2012).  Removal of the soil during mining has resulted in the return to limestone base rock in some areas (Beeton et al., 2010). Additionally, soil fertility is poor as soils are low in organic C, deficient in N, K and S, but extremely high in P (CSBP Report 7QS12022-12038, unpublished) and would be unsuitable for most agricultural crops without considerable inputs of fertilizers.



As has been shown in other degraded areas (Duponnois et al., 2013; Skujinš and Allen, 1986; Thrall et al., 2007), reintroduction of beneficial symbionts such as rhizobia and mycorrhizal fungi, usually absent from severely disturbed soils, can aid in the remediation of degraded soils. Legumes are valuable for revegetation as their symbiotic associations with bacteria and mycorrhizal fungi aids plant growth through nitrogen fixation and P solubilization, respectively (Herrera et al., 1993). The

tropical legume *Mucuna puriens* (L.) DC has been used to increase nitrogen in the soil in phosphate mine rehabilitation in Nauru (Fa'anunu, 2012) and legume species have been shown to be important in the restoration of soil fertility in areas mined for other commodities such as coal (Pallavicini et al., 2015) or degraded through unsustainable agriculture (Hu et al., 2015; Roa-Fuentes et al, 2015)

In many agricultural systems, inoculation using elite strains of rhizobia makes addition of nitrogenous fertilizers unnecessary for legumes and subsequent non-legume crops. In Brazil, the fixation of up to 300 kg N ha$^{-1}$ by soybeans (*Glycine max*) and their bradyrhizobial symbiont, together with the release of 20-30 kg N ha$^{-1}$ into the soils for the following crop saves ~ US\$7 billion a year in nitrogenous fertilizer (Hungria et al., 2013). Soybean represents 50 % of the global crop legume area and 77 % of the N fixed by crop legumes (Herridge et al., 2008). Use of rhizobial inoculants with legumes rather than commercial

nitrogenous fertilizers also contributes to a reduction in greenhouse gas emissions (Hungria et al., 2013).

Microorganisms that enhance plant growth are collectively known as plant growth promoting (PGP) microorganisms, and include symbiotic bacteria, free-living bacteria and mycorrhizal fungi. The beneficial characteristics of PGP microorganisms can include the ability to fix N (Dobbelaere et al., 2003), suppression of plant pathogens and increasing the availability of

poorly soluble plant nutrients (Vessey, 2003). Many free-living PGP bacteria (PGPB) and fungi are able to convert P in the soil to plant-available forms (Richardson et al., 2009). For instance, Panhwar et al., (2011) found that use of two phosphorus solubilizing bacteria (PSB) on aerobic rice significantly improved the phosphorus uptake from Christmas Island rock phosphate resulting in an increase in dry matter yield.

This study aims to establish the scientific basis upon which agriculture can effectively be developed on land that has been previously mined for phosphorus. A research program was developed to improve soil fertility; test the production of high value edible pulses; improve the soils for future agricultural pursuits; and to underpin potential future animal feedlot operations, aquaculture or aquaponic operations. We report here on the use of legumes, and the aid of beneficial microbes, as the first step to improve soil quality for further agricultural production on reclaimed mined areas.

**2 Methodology**

The experiment investigated the response of eight different legume varieties at three fertilizer rates, and to inoculation with rhizobia or co-inoculation with rhizobia and PGPB.



## 2.1 Site preparation

The site (2 ha) had been previously mined for apatite and used as a dump for material after phosphate extraction. The soil form the bunds surrounding the mined areas and the mining debris was bulldozed over the limestone base of the pits, giving a 'soil' depth of just under 1 m. The limestone base was not ripped. As the site had a 5-10 degree slope, three terraces were constructed
to control water run-off. The site surface was scarified with the tynes of a grader and then levelled with the seeder to produce a surface suitable for sowing and harvesting machinery. Soil was collected for analysis by CSBP Limited (Bibra Lake, Western Australia) (Table 1) and then the fertilizer was spread by hand ensuring an even coverage.

## 2.2 Sowing

Seeds for the plant species and their cultivars tested (Table 2) were from plants grown in Australia from commercial suppliers
based in Queensland and passed through the rigorous quarantine requirements for Christmas Island. The rhizobial strains were supplied as commercial peat formulations (ALOSCA Technologies Pty Ltd) (Table 2). The PGPB (*Pseudomonas* sp.) inoculant, an isolate from Western Australian wheatbelt soil that has been shown to produce auxin and siderophores and solubilize phosphorus in vitro (Swift 2016), was prepared by injecting 50 ml of a two-day old broth culture into a sterile peat similar to the rhizobial inoculants. Eight legumes (Table 2) were sown in triplicate and were inoculated with the peat
formulations as rhizobia alone or rhizobia plus the PGPB at three fertilizer rates (Table 3) in a factorial split-plot design. There were 144 machine sown plots and each plot measured 2.5 x 20 m. An additional block (non-replicated) of uninoculated legumes was machine sown on the south-eastern end of the site to allow an assessment of the background rhizobia i.e., nodulation achieved by rhizobia already in the soil. This block received the high level of fertilizer. The soil was dry at the time of sowing (5-9 February 2013), but 20 mm of rain fell within 24 hours and seedling emergence was greater than 90 %.

## 2.3 Assessment

The plants were sampled twice, in April 2013 and in June 2013. At the first sample collection (9 weeks after sowing), ten random plants from each plot (including the uninoculated plots) were dug out ensuring most of the root system remained intact and the nodulation of the roots was assessed using a nodulation rating system (Fig. 1).

Water erosion damage was observed at the first sampling in many plots but particularly the PGPB plots. The damaged areas were also sampled but these plots were removed from statistical analysis. Only one variety was sampled for each of lablab (var. Highworth) and cowpea (var. Ebony) at this assessment. The shoot biomass was removed and dried at 60°C for 2 days and the dry weights determined. At this time, soybean and mungbean had set seed, however, lablab was still in the vegetative stage. Nutrient concentrations in dry matter for the three representative species soybean, mungbean and lablab, were analyzed
by CSBP Limited (Bibra Lake, Western Australia) and the shoot nutrient concentrations were compared with those reported in Reuter and Robinson (1997) at the closest comparable growth stage for each species.

At the second sampling (18 weeks after sowing) the terrace encompassing replicate 3 was completely waterlogged and the biomass was not collected. In the remaining two replicates, the above ground biomass from 1 m$^2$ quadrats placed in areas of average growth in each of the four internal rows of the six rows seeded. Plant material was cut by hand and the fresh weights determined. Representative (10 % total wet biomass) samples were then dried as above to determine dry weights.

## 2.4 Statistical analysis

Means were analysed using a general linear model using IBM SPSS 21 and where significant, were compared using Fisher's least significant difference (LSD) if variance was evenly distributed or Games-Howell analysis if variance was not evenly distributed. Differences were considered to be significant at the $P<0.05$ level.

## 3 Results

### 3.1 Soil analysis

The soil from the experimental site (10 cm) was a clay texture, high in plant available (Colwell) P and low Bo, Cu, K, Mg, N and S (Table 1). The pH was 7.0 (CaCl$_2$) and the low organic C resulted in a low electrical conductivity.

### 3.2 *Vigna unguiculata* (Cowpea)

The application of fertilizer had a significant positive effect on cowpea (var. Ebony) shoot biomass at nine weeks after sowing (Table S1, Fig. 2a). At the low level, fertilizer increased biomass and at the high level there was a further increase compared to the nil fertilizer, in the rhizobia only treatments. At this time, cowpea was nodulated at all the fertilizer levels and in the uninoculated plots, and there was no significant effect of fertilizer or PGPB co-inoculation on nodulation (Table 4).

At 18 weeks after sowing results for the two cowpea varieties were very similar, therefore the data were pooled for analysis (Table S1, Fig. 2b). Again, there was a significant increase in biomass when fertilizer was applied. Co-inoculation with the PGPB did not have a significant effect on plant biomass for cowpea at either of the sampling periods when compared with plants inoculated only with rhizobia.

### 3.3 *Lablab purpureus* (Lablab)

At nine weeks after sowing, the application of fertilizer significantly increased shoot biomass of lablab (var. Highworth) at both fertilizer levels in rhizobia only treatments (Table S1, Fig. 2a). Nodulation of lablab was significantly increased at the low fertilizer level for the rhizobia co-inoculated with PGPB treatments and nodulation score was lower in the uninoculated plots (Table 4).



Comparison of the nutrient analyses of lablab shoots at nine weeks with data from Reuter and Robinson (1997) suggests that copper, potassium (Figure 3b, d) and sulphur (data not shown) were below adequate levels even at the highest rate of fertilizer application. Boron levels were below adequate concentration in plants not co-inoculated with the PGPB (Fig. 3a). Co-inoculation with the PGPB significantly increased the phosphorus concentration in shoots at the nil and low fertilizer applications (Fig. 3c).

The results for the two lablab varieties were very similar at 18 weeks after sowing, therefore the data were pooled (Table S1, Fig. 2b). Fertilizer again had a positive effect on the shoot biomass from the nil to low fertilizer rates for the rhizobia only treatments, however, there was no further increase at the high rate. Co-inoculation with the rhizobia and PGPB significantly decreased lablab biomass at the low fertilizer rate compared with the rhizobia only treatment.

### 3.4 *Vigna radiata* (Mungbean)

Fertilizer application significantly increased the biomass of mungbean 9 weeks after sowing (Table S1, Fig. 2a). Nodulation of mungbean was significantly increased when rhizobia and fertilizer were applied together and the PGPB provided no further advantage (Table 4). Co-inoculation of the mungbean with rhizobia and PGPB had no effect on biomass or shoot nutrient levels at this growth stage. Potassium levels in leaves were well below adequate at all fertilizer levels (Fig. 3d).

At 18 weeks after sowing, coinciding with the end of biomass production, mungbean produced the highest biomass of any of the legume species (Table S1, Fig. 2b). An average dry biomass of 589 g m$^{-2}$ was obtained from plants given low fertilizer rate and co-inoculated with rhizobia and PGPB. The high level of applied fertilizer significantly decreased shoot biomass in the PGPB treatments.

### 3.5 *Phaseolus vulgaris* (Navybean)

The navybean failed to thrive in all treatments and was not collected for shoot analysis at either 9 weeks or 18 weeks after sowing. However, plants were harvested from the uninoculated plots at 9 weeks after sowing to assess nodulation. These plants were as stunted and necrotic as the plants in the inoculated plots, however they were nodulated and received the highest nodule score (4.5) of the legume species sampled from the uninoculated plots (Table 4).

### 3.6 *Cajanus cajan* (Pigeon Pea)

Pigeon pea biomass at nine weeks after sowing was the lowest for all the legume species and fertilizer application did not significantly increase yields (Table S1, Fig. 2a). Co-inoculation with PGPB had no effect and nodulation was relatively low and not significant between the treatments (Table 4). Nodulation was also low in the uninoculated plots.



By 18 weeks after sowing the biomass of pigeon pea was comparable to the other legume species (Table S1, Fig. 2b). Fertilizer application had a significant effect on shoot biomass but this was decreased in the PGPB treatments.

### 3.7 *Glycine max* (Soybean)

Both application of fertilizer and PGPB inoculation had significant positive effects on the shoot biomass of soybean at 9 weeks after sowing (Table S1, Fig. 2a). From the nil to low fertilizer rate, there was a significant increase in biomass and a further increase at the high fertilizer level. At the low fertilizer level, PGPB co-inoculation increased biomass above that gained by fertilizer alone, but no further increase was obtained with PGPB co-inoculation using the high fertilizer rate.

Nodulation of soybean was increased following inoculation with PGPB when fertilizer was not applied (Table 4). Application of fertilizer at the low level also increased nodulation both with and without PGPB inoculation. The uninoculated soybean did not thrive (data not shown) and only two of the collected plants contained nodules.

Fertilizer application significantly increased nutrient concentrations of calcium (not shown), and potassium in soybean shoots, however, potassium levels were still well below the adequate levels determined in Reuter and Robinson (1997) (Figure 3d). Co-inoculation of soybean with rhizobia and PGPB generally increased nutrient concentrations in shoots, however, this was only significantl for zinc (not shown), phosphorus and copper (Figure 3b, c). In fact, copper was below critical deficiency limits without PGPB inoculation at the low fertilizer rate.

Eighteen weeks after sowing, soybean had dropped most of its leaves and pods however the available biomass was harvested and data included. The low rate of fertilizer had a significant positive effect on biomass, but there was no further increase obtained with the high fertilizer rate or co-inoculation with PGPB (Table S1, Figure 2b).

## 4 Discussion

All the legumes (with the exception of navy bean) grew on Christmas Island despite the fact that the soil was highly disturbed, low in nitrogen, and no nitrogen was included with the fertiliser. The application of fertilizer was essential for maximum biomass yields but the response varied among the legume species. In some cases, application of fertilizer at the low level was sufficient to obtain maximum yields. For instance, the highest biomass was from mungbean at low applied fertilizer levels when co-inoculated with rhizobia together with PGPB.

The fertilizer composition used in this study was deliberately broad in order to cater to the requirements of the different legume varieties and the unknown response of the soils to the fertilizer application. However, several essential plant nutrients were below adequate levels in plant tissues of soybean, mungbean and lablab indicating that the fertilizer blend applied to these



species needs to be optimized. The soil is highly disturbed due to the mining activities and is lacking in humic matter and therefore its cation exchange capacity (CEC) is quite low. This was evident by potassium deficiencies in the plants despite the application of 50 kg ha$^{-1}$ potassium sulphate. An increased level of organic carbon in the soil is necessary to improve soil properties (Smith et al., 2015). The continual cropping of legume species will eventually increase the humic matter of the soil

and thereby increase the CEC of the soil, making the exchangeable cations available to subsequent crops. However, until this increase in humic matter is achieved, it is advisable that minerals such as potassium be applied at regular intervals, as a single application may be lost from the soil during the frequent heavy rainfall events. It is essential that fertilizer use is minimized as the soil on Christmas Island is very porous, with rain water reaching the ground water within about six weeks (SGS Economics and Planning Pty. Ltd. and Trust Nature Pty. Ltd. 2010).

Legumes have different requirements for phosphorus, and lablab is better able to utilize phosphorus than other legumes, leading to lower phosphorus requirements (Shehu et al., 2001; Sanginga, 1996). Lablab is common to Africa (Murphy and Colucci 1999) where low phosphorus soils are widespread (Sanginga, 1996) and the inclusion of phosphorus in the fertilizer blends used in the present study appears to be unnecessary given the high level of plant available phosphorus in the Christmas Island

soil. In fact, for the three plant species analysed at nine weeks, phosphorous was in excess in the plant tissues for all treatments. In particular, the lablab treatment (low level fertilizer plus co-inoculated with PGPB) with highest levels of P in shoots had significantly reduced shoot biomass at eighteen weeks. This suggests that phosphorous toxicity and possibly P-induced-Zn deficiency were evident (Hafeez et al., 2013).

Inoculation of legumes with an effective rhizobial strain was vital as demonstrated by the failure of soybean to nodulate with the endemic rhizobia in the uninoculated plots. The other legume species (including navybean) showed some nodulation in the uninoculated plots. The failure of navybean in both the inoculated and uninoculated plots at this site may be due to competition for the commercial inoculant with an ineffective endemic rhizobia. This plant species is highly promiscuous in terms of its ability to form symbioses with a wide range of rhizobia, which in many cases are ineffective and detrimental to

the plant host (Martínez-Romero 2003).

Although no nitrogenous fertilizer was applied and the soil had low nitrogen levels, plant tissue showed adequate nitrogen. Thus for the crop legumes evaluated in this study, inoculation with the commercial strains of rhizobia provided sufficient nitrogen for growth and there was no requirement for expensive nitrogenous fertilizers. This is especially relevant in the

environmentally sensitive environs of Christmas Island.

Soybean and mungbean showed increased growth when co-inoculated with PGPB and rhizobia, and in soybean the PGPB treatments significantly increased the shoot biomass as well as copper, zinc and phosphorous levels in plant tissue. The PGPB used in this study is able to produce siderophores (data not shown) which are known to facilitate iron acquisition by bacteria



and to chelate other metals such as zinc and copper (Braud et al., 2010; Neilands and Leong, 1986). Bacterial siderophores have been shown to effectively provide iron to plants (Radzki et al., 2013) and it is possible that siderophores produced by the PGPB used in the present study were able to provide adequate iron, copper and zinc to the plants.

5    Alternatively, the mechanism by which inoculation with PGPB improves nutrient acquisition in plants could be attributed to alterations in root system architecture, mediated by the synthesis of plant hormones such as auxins and cytokinins by the PGPB, rather than as a direct consequence of nutrient mobilization in soils (Richardson et al., 2009; Richardson and Simpson, 2011). Zamioudis et al., (2013) demonstrated the role of a PGPB in changing the root architecture of *Arabidopsis*. The use of several mutants in signalling pathways implicated the production of molecules with auxin activity as an effector in the 10   alteration of the root system. Therefore, it is also reasonable to suggest that by this mechanism the PGPB used in the present study improved some of the legumes' access to nutrients and water subsequently improving the yields of these plants. Further investigation is required to determine the mechanism(s) of action of this PGPB on these plant species.

The biomass yields achieved in this study were promising for most of the legume species. Selecting cultivars better suited to 15   the seasonal and edaphic conditions on Christmas Island can further increase yields. For instance, the soybean cultivar tested flowered and set seed by April, which was too early to exploit all the seasonal rainfall and a higher yield might be obtained using longer maturing varieties. The dominant abiotic factors influencing soybean phenology are temperature and day length (Setiyono et al., 2007) and the cultivar used in this study flowered and set seed within 8 weeks of sowing. However, with early maturing varieties, it may be possible to achieve two crops per year, potentially increasing productivity even further 20   (Tasma et al., 2011).

As the area of land available for agriculture exceeds that needed for food production for Christmas Island, potential export markets for commercial agriculture to near neighbours such as Malaysia and Indonesia should be considered. For example, in Indonesia there has been a significant decrease in soybean production from 1.87 million tons in 1992 to 0.776 million tons in 25   2008 (Tasma et al., 2011) primarily due to drought (Arumingtyas et al., 2013). Consumption in Indonesia is 2.02 million tons and the deficit is met through imports (Tasma *et al.,* 2011). Cash crops might also include those for fibre, perfumery or medical products.

The farming system developed on Christmas Island must be rain fed. Using ground water during the dry season will not be 30   possible as there is already a heavy draw down on aquifers and salination is a danger (SGS Economics and Planning Pty. Ltd. and Trust Nature Pty. Ltd., 2010) and there are only two wetlands on the island – the Dales and Hosnies Spring. These have high conservation value, being recognized under the Ramsar Convention on wetlands.





During the wet season erosion and temporary waterlogging must be controlled as these caused significant problems in this study. Improving site preparation, possibly ripping of the basal limestone, and runoff control is essential for optimal productivity. The mining exit phases should include consideration of landscaping for subsequent agricultural enterprises. Christmas Island's unique rainforest ecosystems are particularly vulnerable to weed invasion (Director of National Parks, 2010). Past introductions for horticulture or mine site rehabilitation have included some species that have become serious weeds (SGS Economics and Planning Pty. Ltd. and Trust Nature Pty. Ltd., 2010). Undesirable introductions have also caused problems during rehabilitation of other phosphate mines such as in Florida (Brown, 2005). On Christmas Island new plant introductions are strictly controlled by quarantine and several otherwise suitable horticultural species will be impossible to import for this reason.

## 5 Conclusions

The sub-tropical pulse legumes lablab, cowpea, mungbean, pigeon pea and soybean, grew in the disturbed environment on Christmas Island. To optimize yields the selection of appropriate legume cultivars and improvements to site preparation are required. An appropriate rhizobial inoculant for each legume species is essential and the addition of PGPB also increased plant growth. Further studies are needed to determine the ability of the rhizobium and PGPB to persist in these soils and colonise subsequent crops. A fertilizer blend needs to be developed to overcome the deficiencies of potassium, sulphur and copper in the soils and take into account the abundance of phosphorus.

**Author Contributions.** R. Swift, experimental design, data collection and analysis, drafting the manuscript; L. Parkinson, experimental design, data collection and analysis, drafting the manuscript; T. Edwards, data collection and analysis, drafting the manuscript; J. McComb, Drafting and critical revision of the manuscript; R. Carr, data collection and analysis; G. O'Hara, critical revision of the manuscript; G. Hardy, critical revision of the manuscript; L. Bräu, critical revision of the manuscript; J. Howieson, experimental design, data collection, critical revision of the manuscript.

**Acknowledgements.** The authors would like to thank Phosphorus Resources Limited for providing the site, equipment, funding and manpower essential to the execution of this project. We would also like to thank Herve Calmy of Calmy Planning and Design Pty Ltd and Peter Skinner and Neil Ballard of Global Pastures for their invaluable assistance and advice in planning and maintaining this project. We thank Mike Calver for assistance with the statistical analysis and the Australian Government Department of Regional Development and Infrastructure, and Murdoch University for financial support.



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

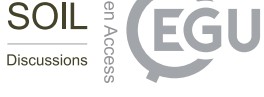

# Tables

**Table 1: Soil analysis for the experimental site (0-10cm).**

| Soil parameter | Unit | Level |
|---|---|---|
| Texture | | Clay |
| Colour | | Brown/orange |
| Ammonium N | mg/kg | 1.00 |
| Nitrate N | mg/kg | 1.00 |
| Colwell P | mg/kg | 299 |
| Colwell K | mg/kg | < 15 |
| S | mg/kg | 2.70 |
| Organic C | % | 0.12 |
| Conductivity | dS/m | 0.04 |
| pH Level (CaCl$_2$) | pH | 7.00 |
| pH Level (H$_2$O) | pH | 7.50 |
| DTPA Cu | mg/kg | 0.82 |
| DTPA Fe | mg/kg | 8.17 |
| DTPA Mn | mg/kg | 12.98 |
| DTPA Zn | mg/kg | 9.04 |
| Exc. Al | meq/100g | 0.03 |
| Exc. Ca | meq/100g | 8.70 |
| Exc. Mg | meq/100g | 0.91 |
| Exc. K | meq/100g | 0.04 |
| Exc. Na | meq/100g | 0.09 |
| Bo Hot CaCl$_2$ | mg/kg | 0.17 |



**Table 2: Legume species used in these experiments and their commercial inoculant.**

| Legume | Species name | Rhizobial strain | Sowing rate (kg/ha) |
|---|---|---|---|
| Soybean | *Glycine max* (L.) Merr. cv. A6785 | *Bradyrhizobium japonicum* CB1809[a] | 60 |
| Mungbean | *Vigna radiata* (L.) R.Wilczek | *Bradyrhizobium* spp. CB1015 | 30 |
| Navybean | *Phaseolus vulgaris* L. | *Rhizobium phaseoli* CC511[b] | 70 |
| Pigeon pea | *Cajanus cajan* Millsp. | *Bradyrhizobium* spp. CB1024 | 30 |
| Lablab | *Lablab purpureus* (L.) Sweet cv. Highworth | *Bradyrhizobium* spp. CB1024 | 30 |
| Lablab | *Lablab purpureus* cv Rongai | *Bradyrhizobium* spp. CB1024 | 30 |
| Cowpea | *Vigna unguiculata* (L.) Walp. cv. Caloona | *Bradyrhizobium* spp. CB1015 | 30 |
| Cowpea | *Vigna unguiculata* cv. Ebony | *Bradyrhizobium* spp. CB1015 | 30 |

[a]CB = The CB *Rhizobium* Collection, Australia
[b]CC = CSIRO Canberra *Rhizobium* Collection, Australia



**Table 3: Fertilizer application rates.**

| Fertilizer rate | Fertilizer applied[a] |
| --- | --- |
| Nil | No fertilizer |
| Low | Potassium sulphate @10 kg/ha |
| | Ferrous sulphate hepta @5.0 kg/ha |
| | Superphosphate[b] @10 kg/ha |
| | TEK Phos 2:1[c]@15 kg/ha |
| High | Five times the low fertilizer rate |

[a]Fertilizers supplied by CSBP Limited

[b]P (w/w %) = 9

5  [c]TEK Phos components (w/w %):  P = 6.0, K = 16.3, S = 6.8, Ca = 12.7, Cu = 0.40, Zn = 0.20, Mo = 0.040



**Table 4:  Mean nodulation score of legume species at 9 weeks after sowing.**

|  | Uninoculated* | Rhizobia only Fertilizer | | | Rhizobia plus PGPB Fertilizer | | |
|---|---|---|---|---|---|---|---|
|  |  | Nil | Low | High | Nil | Low | High |
| Cowpea | 2.4 | 2.8a | 3.9a | 3.4a | 3.7a | 3.1 a | 4.5 a |
| Lablab | 1.6 | 3.7a | 5.0a | 4.0a | 3.8a | 6.1b | 4.0a |
| Mungbean | 3.4 | 3.4a | 5.7b | 5.3b | 4.1ab | 4.4ab | 4.8ab |
| Pigeon pea | 1.4 | 2.1 a | 2.3 a | 2.6 a | 2.6 a | 2.5 a | 2.5 a |
| Soybean | 0.2 | 5.8a | 8.4bc | 7.5abc | 7.2bc | 8.8b | 8.5bc |
| Navybean | 4.5 | na | na | na | na | na | na |

Letters within rows that are the same indicate data that do not differ ($P>0.05$) significantly for the plant species. na, not assessed

*Data from uninoculated plots were not included in the statistical analysis



# Figures

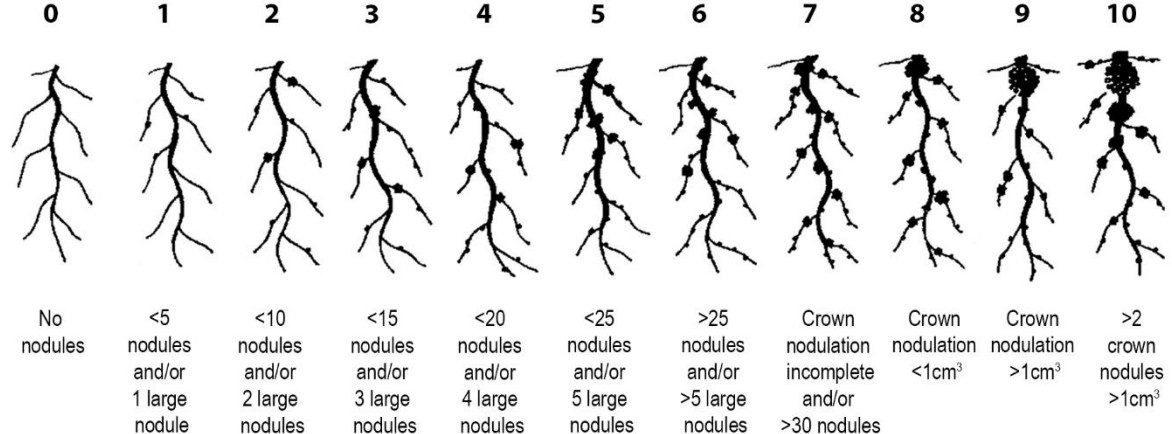

**Figure 1: Nodule scoring system used for scoring pulse legumes. Adequate nodulation can be observed at a nodulation score of 3 and above. Adapted from Howieson and Dilworth (2016).**







**Figure 2: Effect of fertilizer and PGPB on all legume species at (a) 9 weeks after sowing and (b) 18 weeks after sowing. At 9 weeks after sowing, only the varieties Ebony and Highworth were harvested for cowpea and lablab respectively. At 18 weeks after sowing, the results for both cowpea and lablab varieties were pooled. Data are presented as mean shoot dry weight (SDW) and the bars above the columns represent standard error.**



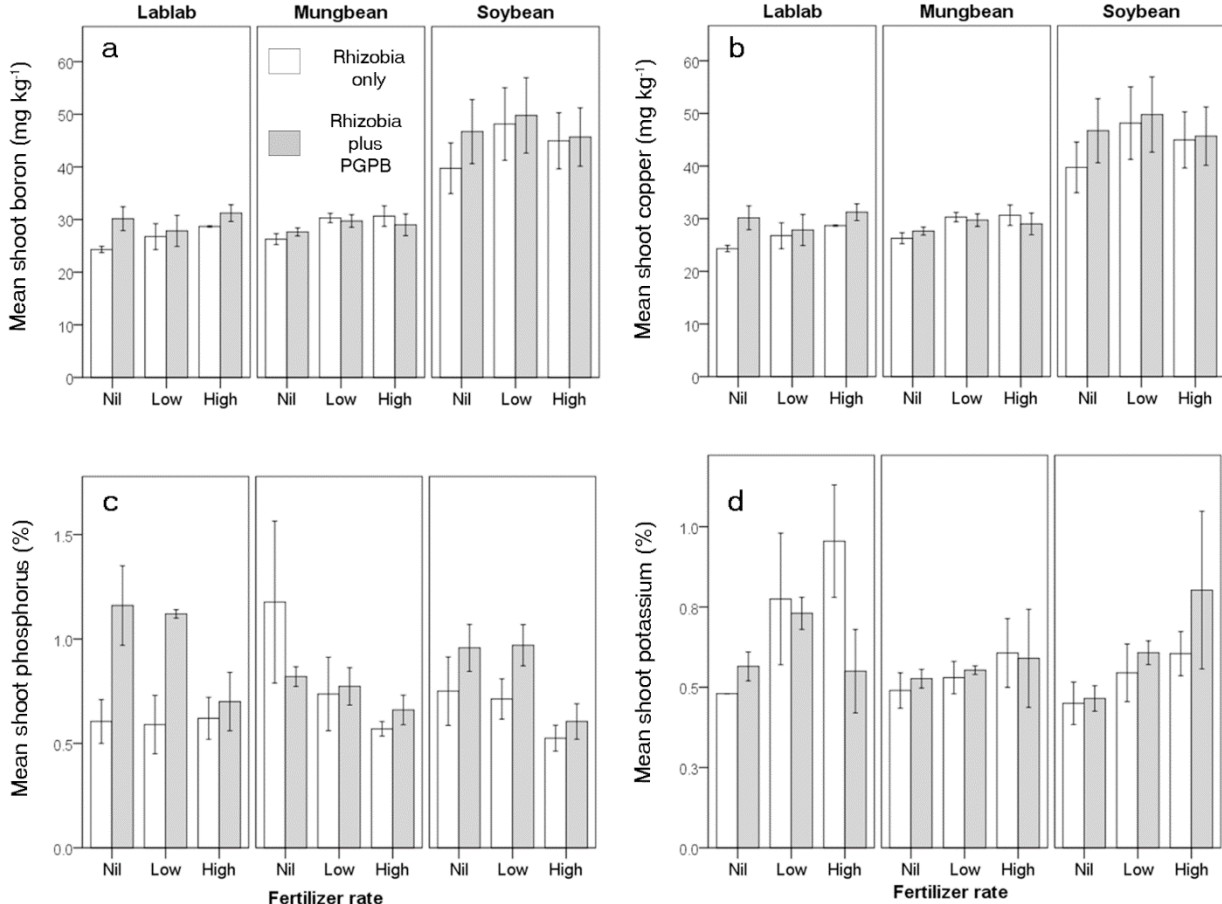

**Figure 3: Concentration of (a) boron (mg kg⁻¹) (b) copper (mg kg⁻¹), (c) phosphorus (%), and (d) potassium (%) in soybean, mungbean and lablab shoots at the nil, low and high rates of fertilizer application, 9 weeks after planting. The bars above the columns represent standard error.**