# Peer review of "Potential for agricultural production on disturbed soils mined for apatite using legumes and beneficial microbes"

_SOIL, 2016_

## Referee Comment (RC1) · Anonymous Referee #1 · 22 Jul 2016

The paper by Swift et al. presents an interesting study that aim to provide a scientific basis to rehabilitate an area previously mined for rock phosphate using six legume species in Christmas Islands. Given the isolation of the area, and the need for developing a sustainable agriculture, this research can provide useful information at the local or regional scale. However the authors should further discuss how this study can be useful at more global scales. Could this particular research be an example or case study of adequate land management for similar areas with analogous status?

The authors claim that rehabilitation of the area with legumes can provide an opportunity to develop alternative industries that would replace mining. However, narrowing the selection of species to restore could have a large impact on biodiversity in the area,

particularly if this occurs at a large scale. Please, elaborate on this issue.

The objective of the study ('establish the scientific basis upon which agriculture can effectively be developed on land that has been previously mined for phosphorus establish' is a little unclear and unspecific. Also, some of these aims are not addressed in the study, e.g. underpin potential future animal feedlot operations, aquaculture or aquaponic operations. Please rewrite considering these comments.

Methods for soil collection and analysis should be explained in detail. Did you collect a bulked composite replicates? Was it replicated? Why did you choose those particular rates of fertiliser? Please explain the rationale for this. Also, why did you add the fertiliser by hand and how did you ensure consistency in the application? Using a non-replicated unfertilised plot can result in an unbalanced design. Please, clarify. The rate of seedling emergence seems quite high (90%). Did you use any treatment to overcome potential dormancy? You affirm that although no nitrogenous fertilizer was applied and the soil had low nitrogen levels, your plant tissue had adequate nitrogen levels. Why do you think you obtained these results?

Other comments/technical corrections: I suggest using colors for the figures. Please, check consistency across the document e.g the use of 'nitrogen' or 'N'; '2 ha' or 'two ha'. Table 1. Explain abbreviations.

---

## Author Comment (AC1) · 4 Aug 2016

Referee's comment should further discuss how this study can be useful at more global scales. Could this particular research be an example or case study of adequate land management for similar areas with analogous status? Response See addition on line 29 pg 3

Referee's comment narrowing the selection of species to restore could have a large impact on biodiversity in the area, particularly if this occurs at a large scale. Please, elaborate on this issue. Response The regions suitable for restoration to rainforest are now mentioned -line 24 pg2-biodiversity is not an issue here

Referee's comment The objective of the study ('establish the scientific basis upon which agriculture can effectively be developed on land that has been previously mined for phosphorus establish' is a little unclear and unspecific.

Response Wording has been changed line 27 pg 3

Referee's comment some of these aims are not addressed in the study, e.g. underpin potential future animal feedlot operations, aquaculture or aquaponic operations. Please rewrite considering these comments. Response Wording has been changed line 28 pg 3.

Referee's comment Why did you choose those particular rates of fertiliser? Please explain the rationale for this.

Response A sentence from the discussion has been shifted to the M&M pg 4 line 10 The fertilizer composition used in this study was deliberately broad in order to cater to the requirements of the different legume varieties and the unknown response of the soils to the fertilizer application.

Referee's comment Using a non- replicated unfertilised plot can result in an unbalanced design. Please, clarify

Response Wording changed line 24 pg 4 to make it clear the plot was fertilised and used only to assess background nodulation.

Referee's comment The rate of seedling emergence seems quite high (90%). Did you use any treatment to overcome potential dormancy?

Response The fact no seed pretreatment was necessary is now mentioned. Pg 4 line 20 The high germination is not unexpected from these agricultural crops.

Referee's comment although no nitrogenous fertilizer was applied and the soil had low nitrogen levels, your plant tissue had adequate nitrogen levels. Why do you think you obtained these results? Response This is adequately covered in the discussion where

we say-the "Although no nitrogenous fertilizer was applied and the soil had low nitrogen levels, plant tissue showed adequate nitrogen. Thus for the crop legumes evaluated in this study, inoculation with the commercial strains of rhizobia provided sufficient nitrogen for growth and there was no requirement for expensive nitrogenous fertilizers. This is especially relevant in the environmentally sensitive environs of Christmas Island."

Referee's comment I would prefer the figures in color

Response Will cost too much

Referee's comment Please, check consistency across the document e.g the use of 'nitrogen' or 'N'; '2 ha' or 'two ha'

Response Have changed chemical symbols to the full word except in places where it is say 30 kg N ha Also changed to two ha in one place

Referee's comment Table 1. Explain abbreviations

Response done

Additional changes

Figure 1 The caption for category 10 has been modified

The data for copper have been corrected

Changes to the text mean that old Fig 3 has become new Fig 2

Please also note the supplement to this comment:
http://www.soil-discuss.net/soil-2016-33/soil-2016-33-AC1-supplement.pdf

[Figure]

| 0 | 1 | 2 | 3 | 4 | 5 | 6 | 7 | 8 | 9 | 10 |
|---|---|---|---|---|---|---|---|---|---|---|
| No nodules | 1-5 nodules and/or 1 large nodule | 6-10 nodules and/or 2 large nodules | 11-15 nodules and/or 3 large nodules | 16-20 nodules and/or 4 large nodules | 21-25 nodules and/or 5 large nodules | 26-30 nodules and/or >5 large nodules | Crown nodulation incomplete and/or >30 nodules | Crown nodulation <1cm$^3$ | Crown nodulation >1cm$^3$ | >1 crown nodule >1cm$^3$ |

**Fig. 1.**

SOILD
[Figure]

**Fig. 2.**

**Supplement:**

**Potential for agricultural production on disturbed soils mined for apatite using legumes and beneficial microbes**

Rebecca Swift[1], Liza Parkinson[1], Thomas J. Edwards, [1] Regina Carr[1], Jen McComb, Graham W. O'Hara[1], Giles E. St. John Hardy[1], Lambert Bräu[1,2], John Howieson[1]

[1]School of Veterinary and Life Sciences, Murdoch University, South Street, Murdoch 6150, Western Australia
[2]Deakin University, Geelong, School of Life and Environmental Sciences, Centre for Regional and Rural Futures, Victoria, Australia
*Correspondence to*: John Howieson (J.Howieson@murdoch.edu.au)

**Abstract.** Christmas Island has been mined for rock phosphate for over 100 years, and as mining will finish in the next few decades there is a need to develop alternative economies on the island, such as high value crop production. However, to conserve the unique flora and fauna on the island, only land previously mined will be considered for this purpose. As these soils have been severely perturbed by mining, strategies to improve soil quality parameters need to be undertaken before plant based industries can be considered. For instance, legumes and beneficial microbes have demonstrated a positive role in the remediation of degraded soils. Therefore, this study aimed to establish the scientific basis upon which agriculture can effectively be developed on soils post phosphate mining. Six legume species (*Glycine max* (Soybean), *Vigna radiata* (Mungbean), *V. unguiculata* (Cowpea), *Phaseolus vulgaris* (Navybean), *Cajanus cajan* Pigeon pea), and *Lablab purpureus* (Lablab) were sown onto a two ha site that had previously been mined for rock phosphate. The soil had a pH of 7.0, and was high in phosphorus but low in boron, copper, potassium, magnesium, nitrogen and sulphur and had low organic carbon. The legumes were inoculated with their respective rhizobial inoculant or co-inoculated with the rhizobia and a plant growth promoting bacteria (PGPB) at three different fertilizer rates (nil, a low rate, and five times the low rate). With the exception of *P. vulgaris*, all the legume species survived. The application of fertilizer was essential for maximum biomass yields 18 weeks after sowing, however the lower fertilizer rate was sufficient to obtain maximum yields for some cultivars. The PGPB increased yields and nodulation of some of the legumes at different fertilizer levels. Although the legumes (except *P. vulgaris*) grew in the Christmas Island environment, selection of appropriate legume cultivars and inoculants plus optimization of the fertilizer regime is required for reliable agricultural productivity on the island

Key words: Christmas Island; rhizobia; Plant Growth Promoting Bacteria; post-mining; rock phosphate

**1 Introduction**

[revised manuscript text omitted]

A major study aims to establish protocols for agricultural production  on land that has been previously mined for phosphorus. A research program was developed to improve soil fertility and test the growth of high value edible pulses, and the results will have application to other areas mined for phospsphate in  the tropics.. We report here on the use of legumes, and the aid of beneficial microbes, as the first step to improve soil quality for further agricultural production on reclaimed mined areas.

**2 Methodology**

The experiment investigated the response of eight different legume varieties at three fertilizer rates, and to inoculation with rhizobia or co-inoculation with rhizobia and PGPB.

**2.1 Site preparation**

5    The site (two ha) had been previously mined for apatite. The soil and mining debris from the bunds surrounding the mined area was bulldozed over the bare limestone of the pits, giving a 'soil' depth of just under 1 m. The limestone base was not ripped. As the site had a 5-10 degree slope, three terraces were constructed to control water run-off. The site surface was scarified with the tynes of a grader and then levelled with the seeder to produce a surface suitable for sowing and harvesting machinery. Soil was collected for analysis by CSBP Limited (Bibra Lake, Western Australia) (Table 1) then the fertilizer

10   was spread by hand ensuring an even coverage. The fertilizer composition used in this study was deliberately broad in order to cater to the requirements of the different legume varieties and the unknown response of the soils to the fertilizer application (Table 2).

**2.2 Sowing**

15   Seeds for the plant species and their cultivars tested (Table 3) were from plants grown in Australia from commercial suppliers based in Queensland and passed through the rigorous quarantine requirements for Christmas Island. The rhizobial strains were supplied as commercial peat formulations (ALOSCA Technologies Pty Ltd) (Table 3). The PGPB (*Pseudomonas* sp.) inoculant, an isolate from Western Australian wheatbelt soil that has been shown to produce auxin and siderophores and solubilize phosphorus in vitro (Swift 2016), was prepared by injecting 50 ml of a two-day old broth culture

20   into a sterile peat similar to the rhizobial inoculants. Eight legumes (Table 3) were sown in triplicate without any seed pretreatment apart from inoculation with the peat formulations for rhizobia alone or rhizobia plus the PGPB at three fertilizer rates (Table 2) in a factorial split-plot design. There were 144 machine sown plots and each plot measured 2.5 x 20 m. An additional block (non-replicated) of uninoculated legumes was machine sown on the south-eastern end of the site. . This block received the high level of fertilizer. The results from the uninoculated block were not included in the statistical

[revised manuscript text omitted]

---

## Referee Comment (RC2) · Anonymous Referee #2 · 13 Aug 2016

The aims and goals behind the paper are commendable and I expect that the paper will ultimately be publishable because its lofty goals and foresight research perspective. However, I feel that the paper would have benefit from a more early intervention to improve structure, table and figures prior to being available as a discussion paper. The paper clearly needs improvement in the organization of the results section, which currently split in 7 subsections some with only 2-4 lines, this give a very fragmented read of this section. It maybe better to have no subheadings at all this also shave at least 7 lines of the paper. The information in Table 1 to 3 is nearly effective captured as text. Table 2 is nearly already describe in the materials and method section anyhow. Figure 1 can go to supplementary material. For Figure 2 and 3 it be interesting to have

the information if differences are significant or not. It not significant maybe a selection can be made of those with are and the others referred to in the text simple as not significant between treatment or species. Clearly, some water damage occurred during the first sampling period, important that this was highlighted and also that damaged area were excluded from the statistical data analysis. However, it is not clear which one or how many species and replicates were in fact removed. In short, the authors should improve the quality of the structure, figure and tables and streamline the main text. Can the focus maybe be more on those species which are actually presented in Fig. 1 and Fig 2. The revised version should be in a much better state for the review of the in depth scientific work described in the paper.

---

## Author Comment (AC2) · 30 Sep 2016

Referee 2 comments and responses It maybe better to have no subheadings (in the Results section) We have reluctantly removed the subheadings

The information in Table 1 to 3 is nearly effective captured as text. Table 2 is nearly already describe in the materials and method section anyhow. We feel Tables 1-3 are core material and do not belong in the supplementary material.

Figure 1 can go to supplementary material. We have shifted the nodule rating figure into the supplementary material. Note also the caption for category 10 has been modified.

For Figure 2 and 3 (now Figs 1 & 2) it be interesting to have the information if differences are significant or not. It not significant maybe a selection can be made of those with are and the others referred to in the text simple as not significant between treatment or species. We mention in the text which treatments are significant. Adding letters to the figures would be messy as the figures are already very busy. The referee seems to suggest using only a few plant species in the figures but this means that we would not be able to compare between weeks 9 and 18 as there are differences we talk about between the 2 different sampling periods. We have included the statistical analysis as a supplementary table for clarification and feel this is better than cluttering up the Figure.

Can the focus maybe be more on those species which are actually presented in Fig. 1 and Fig 2. We feel the emphasis is adequate

, some water damage occurred during the first sampling period, important that this was highlighted and also that damaged area were excluded from the statistical data analysis. However, it is not clear which one or how many species and replicates were in fact removed Now detailed pg 5 line 7

Editors comments Figures better in colour adopted

Insert more recent references References added Bacon and White 2016 Tokar et al. 2016 Gillespie et al. 2015 Yang et al. 2014